# COMPACTER:
# Efficient Low-Rank Hypercomplex Adapter Layers

**Rabeeh Karimi Mahabadi**
EPFL University, Idiap Research Institute
rabeeh.karimi@idiap.ch

**James Henderson**
Idiap Research Institute
james.henderson@idiap.ch

**Sebastian Ruder**
DeepMind
ruder@google.com

## Abstract

Adapting large-scale pretrained language models to downstream tasks via fine-tuning is the standard method for achieving state-of-the-art performance on NLP benchmarks. However, fine-tuning all weights of models with millions or billions of parameters is sample-inefficient, unstable in low-resource settings, and wasteful as it requires storing a separate copy of the model for each task. Recent work has developed *parameter-efficient* fine-tuning methods, but these approaches either still require a relatively large number of parameters or underperform standard fine-tuning. In this work, we propose COMPACTER, a method for fine-tuning large-scale language models with a better trade-off between task performance and the number of trainable parameters than prior work. COMPACTER accomplishes this by building on top of ideas from adapters, low-rank optimization, and parameterized hypercomplex multiplication layers.

Specifically, COMPACTER inserts task-specific weight matrices into a pretrained model's weights, which are computed efficiently as a sum of Kronecker products between shared "slow" weights and "fast" rank-one matrices defined per COMPACTER layer. By only training $0.047\%$ of a pretrained model's parameters, COMPACTER performs on par with standard fine-tuning on GLUE and outperforms standard fine-tuning on SuperGLUE and low-resource settings. Our code is publicly available at https://github.com/rabeehk/compacter.

## 1 Introduction

State-of-the-art pretrained language models (PLMs) in natural language processing (NLP) have used heavily over-parameterized representations consisting of hundreds of millions or billions of parameters to achieve success on a wide range of

> With four parameters I can fit an elephant,
> and with five I can make him wiggle his trunk.
>
> — John von Neumann

NLP benchmarks [2, 3, 4]. These models are generally applied to downstream tasks via fine-tuning [5], which requires updating *all* parameters and storing one copy of the fine-tuned model per task. This causes substantial storage and deployment costs and hinders the applicability of large-scale PLMs to real-world applications. Additionally, fine-tuning of over-parameterized models on low-resource datasets has been shown to be subject to instabilities and may lead to poor performance [6, 7].

Inspired by John von Neumann's quotation, we ask, given that we have already learned general-purpose language representations via a PLM (i.e. we have fit our elephant), how many more parameters

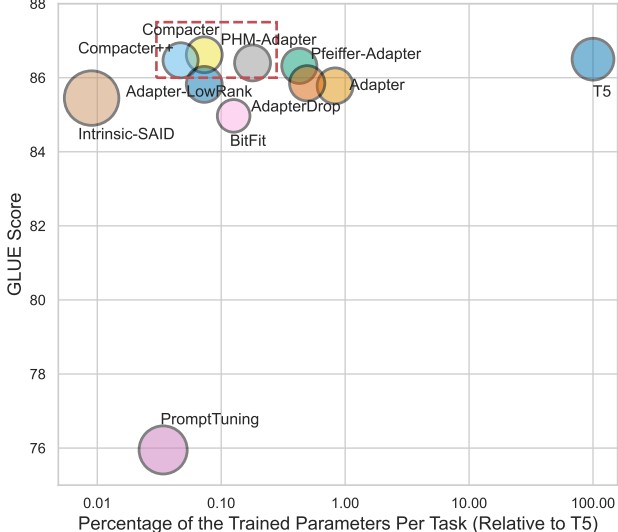

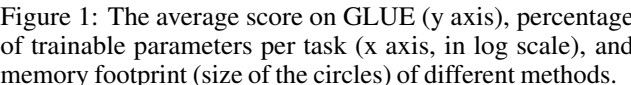

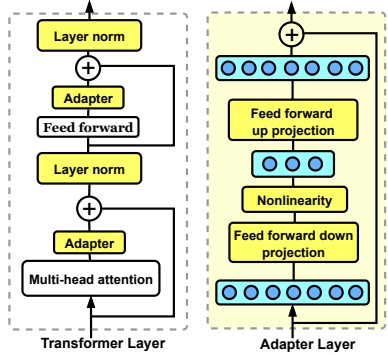

Figure 2: Left: Adapter integration in a pretrained transformer model. Right: Adapter architecture. Following Houlsby et al. [1], we include adapters after the attention and feedforward modules. During training, we only update layer normalizations and adapters (shown in yellow), while the pretrained model is fixed.

Figure 1: The average score on GLUE (y axis), percentage of trainable parameters per task (x axis, in log scale), and memory footprint (size of the circles) of different methods.

do we need to reach state-of-the-art performance on standard NLP tasks. Specifically, we aim to develop practical, memory-efficient methods that train a minimum set of parameters while achieving performance on par or better than full fine-tuning for state-of-the-art NLP models.

Recent literature has introduced *parameter-efficient* fine-tuning methods. These approaches generally keep the pretrained model's parameters fixed and introduce a set of trainable parameters per task, trading off the number of trainable parameters with task performance. At one end of the spectrum, *prompts*, i.e. natural language descriptions of a task, together with demonstrations have been used to achieve reasonable performance *without any* parameter updates on some benchmarks [8] but their performance generally lags behind fine-tuned models. They also require huge models to work well but choosing good prompts becomes harder with larger model sizes [9]. *Soft prompt* methods treat prompts as trainable continuous parameters, which are prepended to the inputs at the input layer or intermediate layers [10, 11, 12]. Such methods, however, often require large models to achieve good performance and are very sensitive to initialization and unstable during training.

The theoretically motivated *low-rank* methods train a small number of parameters that lie in a low-dimensional subspace using random projections [13, 14]. However, storing the random projection matrices causes substantial memory overhead and leads to slow training times. At the other end of the spectrum, *adapter* methods [1, 15] that insert trainable transformations at different layers of the pretrained model require more parameters than the aforementioned approaches but are more memory-efficient and obtain performance comparable to full fine-tuning [1, 16].

In this work, we propose COMPACTER, a method for fine-tuning large-scale language models with an excellent trade-off between the number of trainable parameters, task performance, and memory footprint, compared to existing methods (see Figure 1). COMPACTER builds on ideas from adapters [1], low-rank methods [13], as well as recent hypercomplex multiplication layers [17]. Similar to adapters, COMPACTER inserts task-specific weight matrices into a pretrained model's weights. Each COMPACTER weight matrix is computed as the sum of Kronecker products between shared "slow" weights and "fast" rank-one matrices defined per COMPACTER layer (see Figure 3). As a result, COMPACTER achieves a parameter complexity of $\mathcal{O}(k+d)$ compared to $\mathcal{O}(kd)$ for regular adapters, where the adapters are of size $k \times d$. In practice, COMPACTER trains $0.047\%$ of a PLM's parameters. On the standard GLUE [18] and SuperGLUE [19] benchmarks, COMPACTER outperforms other parameter-efficient fine-tuning methods and obtains performance on par or better than full fine-tuning. On low-resource settings, COMPACTER outperforms standard fine-tuning.

In summary, we make the following contributions: **1)** We propose COMPACTER (**Compact** Adapt**er**) layers, a parameter-efficient method to adapt large-scale language models. **2)** We show that COMPACTER obtains strong empirical performance on GLUE and SuperGLUE. **3)** We demonstrate that

COMPACTER outperforms fine-tuning in low-resource settings. **4)** We provide a parameter complexity analysis of COMPACTER, showing that it requires dramatically fewer parameters than adapters and fine-tuning. **5)** We provide a systematic evaluation of recent parameter-efficient fine-tuning methods in terms of training time and memory consumption. We release our code to facilitate future work.

## 2 Background

We start by introducing the required background on the Kronecker product and adapter layers [1, 15].

### 2.1 Kronecker Product

The Kronecker product between matrix $\boldsymbol{A} \in \mathbb{R}^{m \times f}$ and $\boldsymbol{B} \in \mathbb{R}^{p \times q}$, denoted by $\boldsymbol{A} \otimes \boldsymbol{B} \in \mathbb{R}^{mp \times fq}$, is mathematically defined as:

$$\boldsymbol{A} \otimes \boldsymbol{B} = \begin{pmatrix} a_{11}\boldsymbol{B} & \cdots & a_{1f}\boldsymbol{B} \\ \vdots & \ddots & \vdots \\ a_{m1}\boldsymbol{B} & \cdots & a_{mf}\boldsymbol{B} \end{pmatrix}, \tag{1}$$

where $a_{ij}$ shows the element in the $i^{\text{th}}$ row and $j^{\text{th}}$ column of $\boldsymbol{A}$.

### 2.2 Adapter Layers

Recent work has shown that fine-tuning *all* parameters of a language model can lead to a sub-optimal solution, particularly for low-resource datasets [6]. As an alternative, Rebuffi et al. [15] and Houlsby et al. [1] propose to transfer a model to new tasks by inserting small task-specific modules called *adapter layers* within the layers of a pretrained model, as depicted in Figure 2. They then only train adapters and layer normalizations, while the remaining parameters of the pretrained model remain fixed. This approach allows pretrained language models to efficiently adapt to new tasks.

Each layer of a transformer model is composed of two primary modules: a) an attention block, and b) a feed-forward block. Both modules are followed by a skip connection. As shown in Figure 2, Houlsby et al. [1] suggest to insert an adapter layer after each of these blocks before the skip connection.

Adapters are bottleneck architectures. By keeping the output dimension similar to their input, they cause no change to the structure or parameters of the original model. The adapter layer $A^l$ for layer $l$ consists of a down-projection, $\boldsymbol{D^l} \in \mathbb{R}^{k \times d}$, GeLU non-linearity [20], and up-projection $\boldsymbol{U^l} \in \mathbb{R}^{d \times k}$, where $k$ is the input dimension, and $d$ is the bottleneck dimension for the adapter layer. Adapters are defined as:

$$A^l(\boldsymbol{x}) = \boldsymbol{U^l}(\text{GeLU}(\boldsymbol{D^l}(\boldsymbol{x}))) + \boldsymbol{x}, \tag{2}$$

where $\boldsymbol{x}$ is the input hidden state.

## 3 Method

In this section, we present COMPACTER, a compact and efficient way to adapt large-scale PLMs.

**Problem formulation** We consider the general problem of fine-tuning large-scale language models, where we are given the training data $\mathcal{D} = \{(\boldsymbol{x^i}, y^i)\}_{i=1}^{P}$ with $P$ samples. We assume we are also given a large-scale pretrained language model $f_{\boldsymbol{\theta}}(.)$ parameterized by $\boldsymbol{\theta}$ that computes the output for input $\boldsymbol{x^i}$. Our goal is to fine-tune $f_{\boldsymbol{\theta}}(.)$ efficiently to enable the model to adapt to new tasks.

### 3.1 Compact and Efficient Adapter Layers

In this section, we introduce an efficient version of adapter layers, building on top of recent advances in *parameterized hypercomplex multiplication layers* (PHM) [17]. To the best of our knowledge, we are the first to exploit PHM layers for efficient fine-tuning of large-scale transformer models. The PHM layer has a similar form as a fully-connected layer, which converts an input $\boldsymbol{x} \in \mathbb{R}^k$ to an output $\boldsymbol{y} \in \mathbb{R}^d$:

$$\boldsymbol{y} = \boldsymbol{W}\boldsymbol{x} + \boldsymbol{b}, \tag{3}$$

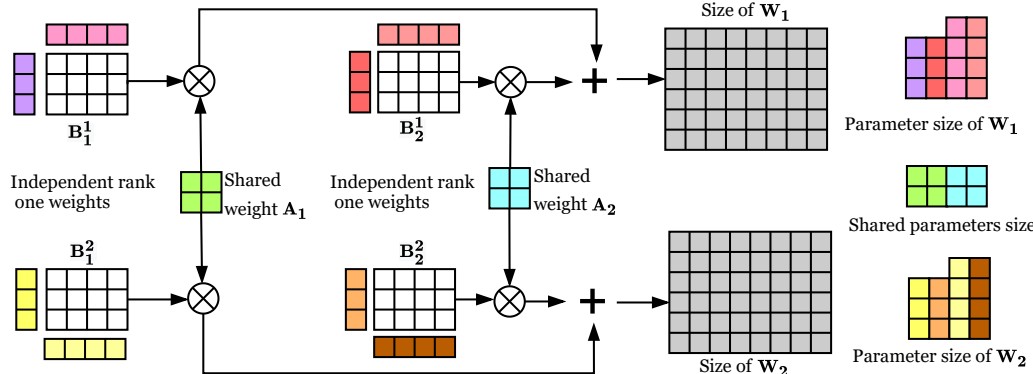

Figure 3: Illustration of generating weights of two different COMPACTER layers: $\boldsymbol{W_1} \in \mathbb{R}^{d \times k}$ (first row) and $\boldsymbol{W_2} \in \mathbb{R}^{d \times k}$ (second row). We generate $\boldsymbol{W_1}$ and $\boldsymbol{W_2}$ using $\boldsymbol{W_j} = \sum_{i=1}^{n} \boldsymbol{A_i} \otimes \boldsymbol{B_i}^{\boldsymbol{j}} = \sum_{i=1}^{n} \boldsymbol{A_i} \otimes (\boldsymbol{s_i}^{\boldsymbol{j}} \boldsymbol{t_i}^{\boldsymbol{j}\top})$ (5), by computing the sum of Kronecker products of *shared* matrices $\boldsymbol{A_i}$ and *adapter-specific* matrices $\boldsymbol{B_i^j}$, with $i \in \{1, ..., n\}$ and adapter index $j \in \{1, 2\}$. We generate each $\boldsymbol{B_i^j}$ by multiplying independent rank one weights. In this example $n = 2$, $d = 6$, and $k = 8$.

where $\boldsymbol{W} \in \mathbb{R}^{k \times d}$. The key difference is that in a PHM layer, $\boldsymbol{W}$ is learned as a sum of Kronecker products. Assume that $k$ and $d$ are both divisible by a user-defined hyperparameter $n \in \mathbb{Z}_{>0}$. Then, the matrix $\boldsymbol{W}$ in (3) is computed as the sum of $n$ Kronecker products as follows:

$$\boldsymbol{W} = \sum_{i=1}^{n} \boldsymbol{A_i} \otimes \boldsymbol{B_i}, \tag{4}$$

where $\boldsymbol{A_i} \in \mathbb{R}^{n \times n}$ and $\boldsymbol{B_i} \in \mathbb{R}^{\frac{k}{n} \times \frac{d}{n}}$. The PHM layer has a parameter complexity of $\mathcal{O}(\frac{kd}{n})$, reducing parameters by at most $\frac{1}{n}$ [17] (see §4).

## 3.2 Beyond Hypercomplex Adapters

Prior work indicates that some of the information captured in pretrained models can be ignored for transfer [21, 22]. Similarly, redundancies have been observed in the information captured by adapters, with adapters in lower layers being less important [1]. In addition, sharing adapters across layers leads to a comparatively small drop of performance for some tasks [23]. Motivated by these insights, we propose the following two extensions to make hypercomplex adapters more efficient.

**Sharing information across adapters**    Sharing all adapter parameters across layers is overall too restrictive and is not able to perform on par with fine-tuning or using regular adapters [23]; however, our decomposition of adapters into $\boldsymbol{A_i}$ and $\boldsymbol{B_i}$ matrices as in Eq. (4) allows us to be more flexible. Consequently, we divide our adaptation weights into *shared* parameters that capture general information useful for adapting to the target task and *adapter-specific* parameters that focus on capturing information relevant for adapting each individual layer. Specifically, we define $\boldsymbol{A_i}$ as shared parameters that are common across all adapter layers while $\boldsymbol{B_i}$ are adapter-specific parameters.

**Low-rank parameterization**    Low-rank methods [13, 14] have demonstrated that strong performance can be achieved by optimizing a task in a low-rank subspace. Similarly, we hypothesize that a model can also be effectively adapted by learning transformations in a low-rank subspace. To this end, we propose to parameterize $\boldsymbol{B_i} \in \mathbb{R}^{\frac{k}{n} \times \frac{d}{n}}$ as a low-rank matrix, which is the product of two low-rank weights $\boldsymbol{s_i} \in \mathbb{R}^{\frac{k}{n} \times r}$ and $\boldsymbol{t_i} \in \mathbb{R}^{r \times \frac{d}{n}}$, where $r$ is the rank of the matrix.[1] Putting both extensions together, we propose the *low-rank* parameterized hypercomplex multiplication layer (LPHM):

$$\boldsymbol{W} = \sum_{i=1}^{n} \boldsymbol{A_i} \otimes \boldsymbol{B_i} = \sum_{i=1}^{n} \boldsymbol{A_i} \otimes (\boldsymbol{s_i} \boldsymbol{t_i}^{\top}). \tag{5}$$

In general, we set $r = 1$ so that $\boldsymbol{B_i}$ is a rank-one matrix. Depending on the complexity of the target task, $r$ can be set to a higher value.[2] Figure 3 illustrates our method. Overall, the LPHM layer reduces

---

[1]We do not factorize $\boldsymbol{A_i}$ as they are small, shared between all layers, and factorization hurts performance.

[2]If factors are over-parameterized, COMPACTER can be used for *overcomplete* knowledge distillation [24].

complexity further to $\mathcal{O}(k+d)$ (see §4). The LPHM layer can also be seen as leveraging "slow" weights $\boldsymbol{A_i}$ that are shared across adapters and capture general information and "fast" weights $\boldsymbol{B_i}$ that learn adapter-specific information for adaptation of each individual layer [25].

**COMPACTER** Based on the above formulation, we introduce COMPACTER layers, which replace the down-projection and up-projection layers in adapters as follows:

$$A^l(\boldsymbol{x}) = \text{LPHM}^{U^l}(\text{GeLU}(\text{LPHM}^{D^l}(\boldsymbol{x}))) + \boldsymbol{x},$$

where the up-projection weights $\text{LPHM}^{U^l}$ are computed as in (5), replacing the layer $U^l$ in (2). Similarly, down-projection weights $\text{LPHM}^{D^l}$ replace the layer $D^l$. While the two adapters in each layer of a transformer have their own $\boldsymbol{s_i}$ and $\boldsymbol{t_i}$ rank-one weights, we share the $\boldsymbol{A_i}$ across all layers and positions of the adapter layers.

## 4 Parameter Efficiency

In this section, we compare the number of parameters of COMPACTER with adapters.

**Adapters parameters** In the standard setting, two adapters are added per layer of a transformer model [1]. Each adapter layer consists of $2kd$ parameters for the down and up-projection matrices ($\boldsymbol{U^l}$, $\boldsymbol{D^l}$) respectively where $k$ is the size of the input dimension and $d$ is the adapter's bottleneck dimension. The total number of parameters for adapters for a transformer model with $L$ layers of both an encoder and a decoder is, therefore, $2L(2kd)$, which scales linearly with all three variables.

**PHM-ADAPTER parameters** In the conventional PHM layer [17], as depicted in Eq. (4), parameters of $\boldsymbol{A_i} \in \mathbb{R}^{n \times n}$ and $\boldsymbol{B_i} \in \mathbb{R}^{\frac{k}{n} \times \frac{d}{n}}$ define the degree of freedom for $\boldsymbol{W}$ as $n(\frac{kd}{n^2} + n^2) = \frac{kd}{n} + n^3$. With the mild condition that $kd > n^4$, then $\frac{kd}{n}$ dominates and the overall parameter size of the PHM layer in (4) is $\mathcal{O}(\frac{kd}{n})$. This condition is satisfied for typical values for adapters, PHM layers, and large-scale PLMs such as T5-large, with hidden size $k = 1024$, adapter hidden size $d \in \{24, 32, 48, 96\}$, and $n = 2, 4, 8, 12$. Hence, the PHM layer offers a parameter reduction of almost $\frac{1}{n}$ compared to standard fully-connected layers, which are $\mathcal{O}(kd)$.[3]

Similarly, employing PHM layers for modeling down and up-projection matrices offers a parameter reduction of almost $\frac{1}{n}$. Each adapter with a PHM layer has in total $2(\frac{kd}{n} + n^3)$ parameters. For a Transformer model with $L$ layers, the total number of parameters of PHM-ADAPTER is $4L(\frac{kd}{n} + n^3)$.

**COMPACTER parameters** COMPACTER shares the trained weight matrices $\{\boldsymbol{A_i}\}_{i=1}^n$ in (5) consisting of $n^3$ parameters across all layers. COMPACTER also has two rank-one weights for each adapter, $\boldsymbol{s_i}, \boldsymbol{t_i}$ in (5) consisting of $\frac{k}{n} + \frac{d}{n}$ parameters, resulting in a total of $2n(\frac{k}{n} + \frac{d}{n})$ parameters for down and up-projection weights. Therefore, the total number of parameters of COMPACTER is $4L(k+d) + n^3$ for a transformer with $L$ layers in the encoder and decoder.

In settings with a large number of layers, the dominant term is $4L(k+d)$. Therefore, with a mild condition that $4L(k+d) > n^3$, COMPACTER has a complexity of $\mathcal{O}(k+d)$, which is far more efficient compared to adapters' $\mathcal{O}(kd)$ and PHM-ADAPTER's $\mathcal{O}(\frac{kd}{n})$ complexity respectively. In settings where $n$ is large, the number of parameters for shared weight matrices $\{\boldsymbol{A_i}\}_{i=1}^n$ for all layers remain constant in COMPACTER with a total of $n^3$ parameters while this scales linearly with the number of layers $L$ for PHM and adapter layers. As an example, in the T5$_{\text{BASE}}$ model with 222M parameters [3], COMPACTER only learns $0.047\%$ of the parameters, and maintains comparable performance to *full fine-tuning*.

## 5 Experiments

**Datasets** Following Raffel et al. [3], we evaluate the performance of the methods on the GLUE [18] and SUPERGLUE [19] benchmarks. These benchmarks cover multiple tasks of paraphrase detection (MRPC, QQP), sentiment classification (SST-2), natural language inference (MNLI, RTE, QNLI, CB), linguistic acceptability (CoLA), question-answering (MultiRC, ReCoRD, BoolQ), word sense disambiguation (WiC), and sentence completion (COPA).[4] As the original test sets are not publicly

---

[3]Even for smaller models where the $n^4$ term dominates, we observe a substantial reduction of parameters compared to adapters.

[4]Following Devlin et al. [2], Raffel et al. [3], as a common practice, we do not experiment with WNLI [26] due to its adversarial nature with respect to the training set.

available, we follow Zhang et al. [27] and split off 1k samples from the training set that we use for validation, while we use the original validation data as the test set. For datasets with fewer than 10k samples (RTE, MRPC, STS-B, CoLA, COPA, WiC, CB, BoolQ, MultiRC), we divide the original validation set in half, using one half for validation and the other for testing.

**Experimental details** We use the state-of-the-art encoder-decoder T5 model [3] as the underlying model for all methods in our experiments. For computational efficiency, we report all results on T5$_{\text{BASE}}$ models (12 encoder and decoder layers and 222M parameters). We use its HuggingFace PyTorch implementation [28]. We fine-tune all methods for 3 epochs on large datasets and 20 epochs for low-resource datasets of GLUE (MRPC, CoLA, STS-B, RTE, BoolQ, CB, COPA, WiC) to allow the models to converge [27]. For all adapter-based methods, we experiment with adapters of bottleneck size of $\{96, 48, 24\}$. We save a checkpoint every epoch for all models and report the results for the hyper-parameters performing the best on the validation set for each task. For the PHM layers, we use the PyTorch implementation of Le et al. [29]. We include low-level details in Appendix A. For our methods, we experiment with $n = \{4, 8, 12\}$ and report the model performing the best. We include the results for all values of $n$ in Appendix B.

Following Mahabadi et al. [30], we freeze the output layer of the pretrained model for all tasks across all methods.[5] We show the results with fine-tuning the output layer in Appendix C. Following Houlsby et al. [1], we update the layer normalization parameters for all methods where applicable.[6]

## 5.1 Baselines

We compare against several recently proposed *parameter-efficient* fine-tuning methods:

**T5$_{\text{BASE}}$** We compare our method to the standard practice of fine-tuning T5, where we fine-tune all parameters of the model on each individual task.

**ADAPTER** We compare to a strong adapter baseline [1], which adds adapters for each task after the feed-forward and attention modules in each transformer block of T5.

**PFEIFFER-ADAPTER** Pfeiffer et al. [31] propose a more efficient adapter variant, which keeps only one of the adapters in each layer for better training efficiency. We experimented with keeping either adapter and found keeping the adapter after the self-attention module in each layer to perform the best.

**ADAPTER-LOWRANK** We parameterize each adapter's weight as a product of two rank-one weights.

**PROMPT TUNING** Prompt tuning [12] is the successor variant of Li and Liang [10], which prepends a randomly initialized continuous prompt to the input (PROMPT TUNING-R). We also compare to a variant, which initializes prompts using token embeddings of the pretrained language model's vocabulary (PROMPT TUNING-T) [12].

**INTRINSIC-SAID** The Structure Aware Intrinsic Dimension [14] fine-tunes the model by reparameterizing the parameters in a lower-dimensional subspace $\theta^{d'}$ ($d' \ll D$): $\theta_i^D = \theta_{i,0}^D + \lambda_i P \theta_i^{d'-m}$ where parameter $\theta_{i,0}^D$ are the pretrained model's parameters and $P \in \mathbb{R}^{d'-m} \to \mathbb{R}^D$ is a random linear projection via the Fastfood transform [32]. They then consider the total number of weight matrices in the PLM, $m$, and attribute a weight to each of them, resulting in $\lambda \in \mathbb{R}^m$ in total by trading $m$ parameters from the low dimensional space $\theta^{d'} \in \mathbb{R}^{d'}$. Then, the total trainable parameters are $\theta^{d'-m} \in \mathbb{R}^{d'-m}$ and $\lambda$.

**ADAPTERDROP** We apply the method of Rücklé et al. [23], which drops the adapters from lower transformer layers for a better training efficiency to T5 with ADAPTER. Consequently, we drop adapters from the first five layers of both the encoder and the decoder in T5$_{\text{BASE}}$.

**BITFIT** Cai et al. [33] propose to freeze the weights and only train the biases. By not storing intermediate activations, this method enables substantial memory savings. Ravfogel et al. [34] study a similar method for PLMs that fine-tunes only the biases and the final output layer.[7]

---

[5]This is much more efficient as the output layer includes 11.1% of the parameters of T5$_{\text{BASE}}$. Tasks are formulated in a text-to-text format so the model can be applied to them without learning a new output layer [3]. We note that this is in contrast to the original adapter setting, which used an encoder-only masked PLM [1].

[6]For BITFIT, we only update the biases. For PROMPT TUNING, the entire model is frozen.

[7]Note that in the HuggingFace T5 implementation, the biases in layer normalizations, linear layers, the output layer and self-attention layers are removed. We re-introduce these biases for BITFIT.

Table 1: Performance of all models on the GLUE tasks. For each method, we report the total number of parameters across all tasks and the number of parameters that are trained for each task as a multiple and proportion of $T5_{BASE}$ model [3]. For MNLI, we report accuracy on the matched validation set. For MRPC and QQP, we report accuracy and F1. For STS-B, we report Pearson and Spearman correlation coefficients. For CoLA, we report Matthews correlation. For all other tasks, we report accuracy. Bold fonts indicate the best results. For the results with †, due to insatiability during training, we restarted experiments with 6 random seeds and report the best. For INTRINSIC-SAID, $d'$ is set to 20K.

| Method | #Total params | Trained params / per task | CoLA | SST-2 | MRPC | QQP | STS-B | MNLI | QNLI | RTE | Avg |
|---|---|---|---|---|---|---|---|---|---|---|---|
| *Baselines* | | | | | | | | | | | |
| T5$_{BASE}$ | 8.0×1 | 100% | 61.76 | **94.61** | **90.20/93.06** | **91.63/88.84** | 89.68/89.97 | **86.78** | 93.01 | 71.94 | **86.50** |
| ADAPTER | 1.065 | 0.832% | **64.02** | 93.81 | 85.29/89.73 | 90.18/87.20 | 90.73/91.02 | 86.49 | 93.21 | 71.94 | 85.78 |
| PFEIFFER-ADAPTER | 1.032 | 0.427% | 62.9 | 93.46 | 86.76/90.85 | 90.14/87.15 | 91.13/91.34 | 86.26 | 93.30 | **76.26** | 86.32 |
| ADAPTERDROP | 1.038 | 0.494% | 62.7 | 93.58 | 86.27/90.60 | 90.2/87.25 | **91.37/91.61** | 86.27 | 93.23 | 71.22 | 85.85 |
| ADAPTER-LOWRANK | 1.004 | 0.073% | 59.19 | 93.69 | 88.24/91.49 | 90.23/87.01 | 90.8/91.33 | 85.8 | 92.9 | 73.38 | 85.82 |
| PROMPT TUNING-R | 1.003 | 0.034% | 0.47† | 87.61 | 68.14/81.05 | 88.93/85.55 | 90.25/90.59 | 46.83† | 92.33 | 54.68 | 71.49 |
| PROMPT TUNING-T | 1.003 | 0.034% | 10.59 | 90.94 | 68.14/81.05 | 89.69/86.14 | 89.84/90.21 | 81.46 | 92.75 | 54.68 | 75.95 |
| INTRINSIC-SAID | 1.001 | 0.009% | 58.69 | 94.15 | 88.24/91.78 | 90.28/87.13 | 90.06/90.45 | 85.23 | **93.39** | 70.50 | 85.45 |
| BITFIT | 1.010 | 0.126% | 58.16 | 94.15 | 86.76/90.53 | 90.06/86.99 | 90.88/91.26 | 85.31 | 92.99 | 67.63 | 84.97 |
| *Our Proposed Methods* | | | | | | | | | | | |
| PHM-ADAPTER ($n=12$) | 1.013 | 0.179% | 57.35 | 94.50 | 91.67/93.86 | 90.25/87.05 | 90.45/90.84 | **85.97** | 92.92 | 75.54 | 86.40 |
| COMPACTER ($n=4$) | 1.004 | 0.073% | **63.75** | 93.00 | 89.22/92.31 | 90.23/87.03 | 90.31/90.74 | 85.61 | 92.88 | **77.70** | **86.62** |
| COMPACTER++ ($n=4$) | 1.002 | 0.047% | 61.27 | 93.81 | 90.69/93.33 | 90.17/86.93 | **90.46/90.93** | 85.71 | **93.08** | 74.82 | 86.47 |

## 5.2 Our Methods

**PHM-ADAPTER** We learn the weights of adapters using PHM layers as in (4). To our knowledge, we are the first who exploit the idea of PHM [17] for efficient *fine-tuning* of large-scale language models.

**COMPACTER** We learn adapter weights using LPHM layers as described in (5). We also explore a variant where we only keep the COMPACTER layer after the feed-forward layer in each transformer block (COMPACTER++).[8]

## 5.3 Results on the GLUE Benchmark

Table 1 shows the results on GLUE with $T5_{BASE}$ (see Appendix E for results on $T5_{SMALL}$). COMPACTER and COMPACTER++ outperform all previous parameter-efficient methods and perform on par with full fine-tuning while only training 0.07% and 0.047% of parameters respectively. We now discuss the different methods in detail.

**Adapter-based methods** For ADAPTER, not fine-tuning the classifier hurts the performance substantially (85.78 versus 86.48; cf. Appendix C). PFEIFFER-ADAPTER, which adds adapters only after the self-attention module outperforms the standard ADAPTER while being more parameter-efficient. ADAPTERDROP obtains lower performance than fine-tuning, demonstrating that adapting the lower layers of an encoder-decoder T5 model is important for its performance. Additionally, ADAPTER-LOWRANK is not expressive enough to perform well on this benchmark.

**Prompt tuning and BitFit** For PROMPT TUNING, we observe high sensitivity to initialization and learning rate, as also confirmed in [10]. We experimented with multiple random seeds but performance lags behind fine-tuning substantially, in particular on low-resource datasets. This can be explained by the low flexibility of such methods as all the information needs to be contained in the prefixes. As a result, the method only allows limited interaction with the rest of the model and good performance requires very large models [12]. In addition, increasing the sequence length leads to memory overhead (see §5.5) and the number of prompt tokens is limited by the number of tokens that can fit in the model's maximum input length, which makes such methods less flexible and unsuitable for dealing with large contexts. Similarly, BITFIT performs worse than fine-tuning, especially on low-resource datasets.

**Intrinsic-SAID** Interestingly, the average performance of INTRINSIC-SAID, which fine-tunes only 0.009% of a model's parameters is only 1.05 points below the fine-tuning baseline. However, this method has two practical drawbacks: a) storing the random projection matrices results in a substantial

---

[8]We found this to slightly outperform keeping the COMPACTER layer after the self-attention layer instead.

Table 2: Performance of all methods on the SUPERGLUE tasks. For each method, we report the total number of parameters across all tasks and the percentage of parameters that are trained for each task as a multiple and proportion of T5$_{\text{BASE}}$ model [3]. For CB, we report accuracy and F1. For MultiRC, we report F1 over all answer-options (F1$_a$) and exact match of each question's set of answers (EM) [19]. For ReCoRD, we report F1 and EM scores. For all other tasks, we report accuracy. For INTRINSIC-SAID, $d'$ is set to 20K. Bold fonts indicate the best results in each block.

| Method | #Total params | Trained params / per task | BoolQ | CB | COPA | MultiRC | ReCoRD | WiC | Avg |
|---|---|---|---|---|---|---|---|---|---|
| *Baselines* | | | | | | | | | |
| T5$_{\text{BASE}}$ | 6.0×1 | 100% | 81.10 | **85.71/78.21** | 52.0 | 68.71/47.0 | 74.26/73.33 | **70.22** | 70.06 |
| ADAPTER | 1.049 | 0.832% | 82.39 | 85.71/73.52 | 52.0 | 72.75/53.41 | 74.55/73.58 | 67.08 | 70.55 |
| PFEIFFER-ADAPTER | 1.024 | 0.427% | **82.45** | 85.71/75.63 | 54.0 | 72.53/51.76 | 74.69/73.70 | 68.65 | **71.01** |
| ADAPTERDROP | 1.028 | 0.494% | 82.26 | 85.71/75.63 | 42.0 | **72.92/53.30** | 74.68/73.70 | 68.34 | 69.84 |
| ADAPTER-LOWRANK | 1.003 | 0.073% | 80.31 | 78.57/55.37 | 54.0 | 72.58/51.98 | 74.77/73.87 | 64.58 | 67.34 |
| PROMPT TUNING-R | 1.002 | 0.034% | 61.71 | 67.86/46.99 | 48.0 | 59.23/16.33 | 75.27/74.36 | 48.90 | 55.41 |
| PROMPT TUNING-T | 1.002 | 0.034% | 61.71 | 67.86/46.89 | 52.0 | 57.66/19.44 | **75.37/74.41** | 48.90 | 56.03 |
| INTRINSIC-SAID | 1.001 | 0.009% | 78.72 | 75.00/51.83 | 54.0 | 69.98/52.78 | 74.86/73.91 | 65.83 | 66.32 |
| BITFIT | 1.008 | 0.126% | 79.57 | 78.57/54.40 | **56.0** | 70.73/48.57 | 74.64/73.64 | 69.59 | 67.30 |
| *Our Proposed Methods* | | | | | | | | | |
| PHM-ADAPTER ($n=4$) | 1.013 | 0.240% | **80.31** | 85.71/73.52 | 44.0 | **71.99/51.65** | 74.62/73.60 | 67.40 | 69.20 |
| COMPACTER ($n=12$) | 1.003 | 0.073% | 78.59 | **96.43/87.44** | 48.0 | 70.80/49.67 | 74.49/73.54 | 65.20 | 71.57 |
| COMPACTER++ ($n=12$) | 1.002 | 0.048% | 78.84 | 92.86/84.96 | **52.0** | 70.68/50.99 | 74.55/73.50 | **68.03** | **71.82** |

memory overhead; b) it is very slow to train (see §5.5). Despite this, INTRINSIC-SAID provides insights regarding the effectiveness of low-rank optimization of pretrained language models [14], which motivates the development of parameter-efficient methods such as COMPACTER.

**COMPACTER** For our proposed methods, we observe fine-tuning the output layer for both PHM-ADAPTER and COMPACTER++ does not provide much performance difference (see Appendix C). PHM-ADAPTER reduces the parameters of ADAPTER from 0.83% to 0.179% (with $n=12$), being $4.64\times$ more parameter-efficient. COMPACTER reduces the number of parameters to the remarkable rate of 0.073% while obtaining comparable results to full fine-tuning. By removing the COMPACTER layer after self-attention, COMPACTER++ obtains similar performance, while reducing the parameters to 0.047%. Adaptation without updating the layer normalization can be a promising direction to reduce the parameters further, for instance by building on recent advances in normalization-free models [35], which we leave to future work.

## 5.4 Results on the SUPERGLUE Benchmark

Table 2 shows the performance of the methods on SUPERGLUE [19]. We include the results for all values of $n$ in Appendix D. We observe a similar pattern as on GLUE in Table 1. COMPACTER and COMPACTER++ perform substantially better compared to other parameter-efficient fine-tuning methods and even outperform full fine-tuning while only training 0.073% and 0.048% of the parameters.

## 5.5 Efficiency Evaluation

In this section, we compare the efficiency of our proposed methods with various recently proposed parameter-compact fine-tuning methods under the same computation budget. To this end, we train all methods for 1 epoch on the MNLI dataset. For each method, we select the largest batch size that fits a fixed budget of the GPU memory (24 GB). For all adapter-based methods, we fix the adapter size to 24. For PROMPT TUNING, we set the number of prefix tokens to 100. For INTRINSIC-SAID, we set $d' = 1400$. Finally, we set $n = 4$. In Table 3, we report the percentage of trained parameters per task, training time per epoch, and memory usage of each method. Moreover, Figure 1 shows the trade-off between quantitative performance, percentage of trained parameters, and memory footprint.

Our approaches have several attractive properties. Based on our analysis in Table 1, COMPACTER and COMPACTER++ obtain the best combination of high GLUE score averaged across all tasks, plus a substantially lower number of parameters (0.073% and 0.047% respectively). In addition to COMPACTER++

Table 3: Percentage of trained parameters per task, average peak memory and training time for all methods. $\Delta\%$ is the relative difference with respect to *full fine-tuning* (T5$_{\text{BASE}}$). Lower is better.

| Method | Trained params/ per task | Memory (MB) | $\Delta\%$ | Time/Epoch (min) | $\Delta\%$ |
|---|---|---|---|---|---|
| T5$_{\text{BASE}}$ | 100% | 167.99 | — | 42.13 | — |
| ADAPTER | 0.832% | 124.02 | -35.45% | 31.81 | -24.50% |
| PFEIFFER-ADAPTER | 0.427% | 118.4 | -41.88% | 28.19 | -33.09% |
| ADAPTERDROP | 0.494% | 119.41 | -40.68% | 28.08 | -33.35% |
| ADAPTER-LOWRANK | 0.073% | 123.8 | -35.69% | 32.71 | -22.36% |
| PROMPT TUNING | 0.034% | 222.27 | 24.42% | 44.54 | 5.72% |
| INTRINSIC-SAID | 0.009% | 285.40 | 41.14% | 144.01 | 241.82% |
| BITFIT | 0.126% | 102.31 | -64.20% | 27.36 | -35.06% |
| PHM-ADAPTER | 0.179% | 123.93 | -35.55% | 35.55 | -15.62% |
| COMPACTER | 0.073% | 123.91 | -35.57% | 36.48 | -13.41% |
| COMPACTER++ | 0.047% | 118.35 | -41.94% | 30.96 | -26.51% |

performing well, its memory requirement is the second best among all methods, reducing memory usage by -41.94% compared to T5$_{\text{BASE}}$. COMPACTER and COMPACTER++ also speed up training substantially, by -13.41% and -26.51% relative to T5$_{\text{BASE}}$. On the other hand, BITFIT, by not storing intermediate activations, has the lowest memory requirement (-64.2% relative to T5$_{\text{BASE}}$) and is the fastest (-35.06% relative to T5$_{\text{BASE}}$) at the cost of lower quantitative performance (1.53 points lower; see Table 1).

Methods relying on pruning adapters, i.e., PFEIFFER-ADAPTER and ADAPTERDROP reduce the memory overhead and improve training time. However, their number of parameters is almost an order of magnitude more compared to COMPACTER++, with 9.1× and 10.5× more parameters respectively. Moreover, although, PFEIFFER-ADAPTER performs on par with full fine-tuning with a slight degradation (Table 1), ADAPTERDROP obtains a lower performance (-0.65 less on average across all tasks.). We note that dropping adapters from transformer layers is a general technique and could be applied to COMPACTER for improving efficiency even further, which we leave to future work. Similarly, although ADAPTER-LOWRANK reduces the memory overhead and improves the training time, it obtains a lower performance (Table 1) (-0.68 less on average across all tasks.).

At the other end of the spectrum, INTRINSIC-SAID and PROMPT TUNING methods have the lowest number of parameters. However, they both come with high memory overhead (41.14% and 24.42% relative to full fine-tuning (T5$_{\text{BASE}}$) respectively), are slowest to train, and their performance substantially lags behind full fine-tuning (see Table 1). For PROMPT TUNING, high memory costs are due to the fact that the computational complexity of self-attention, which requires storing the full attention matrix for gradient computation, scales quadratically with the sequence length [36]. For INTRINSIC-SAID, the high memory requirement is due to storing large random projection matrices, which limits the application of INTRINSIC-SAID for fine-tuning large-scale PLMs. Moreover, computing projections via FastFood transform, although theoretically possible in $O(D\log d')$ [32], is slow in practice even with a CUDA implementation. For pretrained language models with a large number of parameters, allocating random projections for the full parameter space is intractable. While using Fastfood transform partially ameliorates this issue by reducing the memory usage from $\mathcal{O}(Dd')$ to $\mathcal{O}(D)$, the memory issue with such methods remains unresolved.

Overall, given the size of large-scale transformer models with millions and billions of parameters, such as T5 [3], efficient memory usage is of paramount importance for practical applications. COMPACTER and COMPACTER++ offer a great trade-off in terms of performance, memory usage, and training time. With regard to our inspiration of von Neumann's quotation, we thus find that only a comparatively small number of additional parameters are necessary for the practical and efficient adaptation of PLMs.

## 5.6  Low-resource Fine-tuning

COMPACTER++ has substantially fewer parameters compared to T5$_{\text{BASE}}$. In this section, we investigate whether this could help COMPACTER++ to generalize better in resource-limited settings. We subsample each dataset of GLUE for varying sizes in the range $\{100, 500, 1000, 2000, 4000\}$. Figure 4 shows the

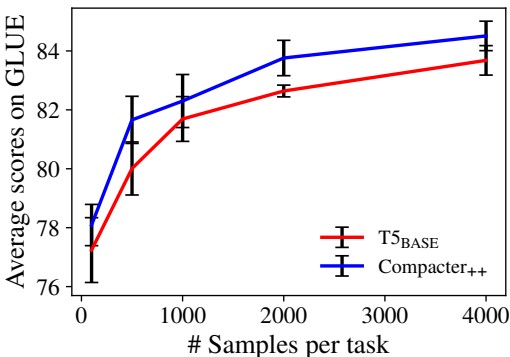

Figure 4: Results on GLUE for the various number of training samples per task (100,500,1000,2000,4000). We show mean and standard deviation across 5 seeds.

results. COMPACTER++ substantially improves the results in the low-resource setting, indicating more effective fine-tuning in this regime.

## 6 Related Work

**Adapters** Adapters have recently emerged as a new paradigm for fine-tuning pretrained language models [1]. In another line of work, Üstün et al. [37] proposed a multilingual dependency parsing method based on adapters and contextual parameter generator networks [38], where they generate adapter parameters conditioned on trained input language embeddings. This, however, leads to a large number of additional parameters compared to the base model. Contemporaneously, Mahabadi et al. [30] use a single compact hypernetwork allowing to generate adapter weights efficiently conditioned on multiple tasks and layers of a transformer model. Pilault et al. [39] also proposed a task-conditioned transformer for multi-task learning which is less parameter-efficient. The aforementioned work is complementary to COMPACTER, and one could potentially combine COMPACTER with contextual parameter generation to generate adapter modules. Compared to Mahabadi et al. [30], COMPACTER++ reduces the parameters by $6.2\times$.

**Hypercomplex representations** Deep learning advances in the hypercomplex domain are in a nascent stage, and most work is fairly recent [40, 41, 42, 43, 44]. Replacing matrix multiplications in standard networks with Hamilton products that have fewer degrees of freedom offers up to a $4\times$ saving of parameter size in a single multiplication operation [42, 44]. Very recently, Zhang et al. [17] extend such methods in a way that they could reduce the parameters of a fully connected layer under a mild condition to $1/n$, where $n$ is a user-specified parameter. To the best of our knowledge, there is no previous work that attempts to leverage the hypercomplex space for efficient fine-tuning of large-scale language models.

**Other parameter-efficient models** Li et al. [13] and Aghajanyan et al. [14] study training models in a low-dimensional randomly oriented subspace instead of their original parameter space. Another recent line of work has shown that pretrained models such as BERT are redundant in their capacity, allowing for significant sparsification without much degradation in end metrics [45, 46, 47]. Such methods, however, remain not well supported by current hardware and often perform worse compared to dedicated efficient architectures [48].

## 7 Conclusion

We have proposed COMPACTER, a light-weight fine-tuning method for large-scale language models. COMPACTER generates weights by summing Kronecker products between shared "slow" weights and "fast" rank-one matrices, specific to each COMPACTER layer. Leveraging this formulation, COMPACTER reduces the number of parameters in adapters substantially from $\mathcal{O}(kd)$ to $\mathcal{O}(k+d)$. Through extensive experiments, we demonstrate that despite learning $2127.66\times$ fewer parameters than standard fine-tuning, COMPACTER obtains comparable or better performance in a full-data setting and outperforms fine-tuning in data-limited scenarios.

## Acknowledgements

We are grateful to Dani Yogatama for feedback on a draft of this manuscript. The authors would like to thank Tuan Le for his assistance in reproducing the results of Zhang et al. [17]. We would like to also thank Armen Aghajanyan for his assistance to reproduce the results of his work [14]. We thank Jue Wang for his comments on an earlier version of this paper. The authors are grateful to Brian Lester, Rami Al-Rfou, Noah Constant, and Mostafa Dehghani for their assistance. Rabeeh Karimi Mahabadi was supported by the Swiss National Science Foundation under the project Learning Representations of Abstraction for Opinion Summarization (LAOS), grant number "FNS-30216".

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
