Table 4: Selected learning rates for all methods.

| Method | Learning rate |
|---|---|
| T5$_{\text{BASE}}$ | 3e−4 |
| ADAPTER | 3e−4 |
| PFEIFFER-ADAPTER | 3e−4 |
| ADAPTERDROP | 3e−4 |
| ADAPTER-LOWRANK | 3e−3 |
| PROMPT TUNING-R | 3e−2 |
| PROMPT TUNING-T | 3e−1 |
| INTRINSIC-SAID | 3e−2 |
| BITFIT | 3e−4 |
| PHM-ADAPTER | 3e−3 |
| COMPACTER | 3e−3 |
| COMPACTER++ | 3e−3 |

Table 5: Selected learning rates for all methods, when we also fine-tune the output layer.

| Method | Learning rate |
|---|---|
| ADAPTER | 3e−3 |
| PFEIFFER-ADAPTER | 3e−4 |
| ADAPTERDROP | 3e−4 |
| ADAPTER-LOWRANK | 3e−3 |
| BITFIT | 3e−4 |
| PHM-ADAPTER | 3e−3 |
| COMPACTER | 3e−3 |
| COMPACTER++ | 3e−3 |

## A Experimental Details

**Datasets**   We run all experiments on the standard GLUE benchmark [18] with Creative Commons license (CC BY 4.0) and the SUPERGLUE benchmark [19]. These benchmark consist of multiple datasets: CoLA [49], SST-2 [50], MRPC [51], QQP[9], STS-B [52], MNLI [53], QNLI [54], and RTE, which is a combination of data from RTE1 [55], RTE2 [56], RTE3 [57], RTE5 [58], COPA [59], CB [60], MultiRC [61], ReCoRD [62], BoolQ [63], and WiC [64] where sentences are selected from VerbNet [65], WordNet [66], and Wiktionary. We download all datasets from the HuggingFace Datasets library [67].

**Low-resource fine-tuning**   For the experiment conducted in §5.6, we set the number of epochs to 1000, 200, 100, 50, 25, for datasets subsampled to size 100, 500, 1000, 2000, and 4000 respectively. Based on our results, this is sufficient to allow the models to converge. We save a checkpoint every 250 steps for all models and report the results for the hyper-parameters performing the best on the validation set for each task.

**Data pre-processing:**   Following Raffel et al. [3], we cast all datasets into a sequence-to-sequence format. We recast STS-B as a 21-class classification task by rounding its target scores to their nearest increment of 0.2.

**Computing infrastructure:**   We run the experiments in Table 1, 2, 9, and 3 on one NVIDIA GEFORCE RTX 3090, and experiments in §5.6 on one GEFORCE GTX 1080 TI GPU.

**Training hyper-parameters:**   For the experiments on GLUE, we set the maximum sequence length to 128 and batch size to 100. Following Raffel et al. [3], we use maximum sequence length of 256 for the tasks in SUPERGLUE, and for ReCoRD, we set it to 512. We used batch size of 32 for SUPERGLUE, and for ReCoRD, we set it to 16 due to the GPU memory limit. For results in §5.6, we set the batch size to 40 to match the lower GPU memory of GEFORCE GTX 1080 TI GPU. For setting the learning rates, we trained all methods with 3e−5, 3e−4, 3e−3, 3e−2, and 3e−1 and use the learning rate performing the best on the validation set for each method. Table 4 shows the final selected learning rate for each method reported in Table 1. For the method variants where we also fine-tune the final output layer (Table 7), we report the selected learning rate in Table 5. We train all models with the AdamW optimizer from the HuggingFace library [28] with default hyper-parameters of $\beta_1 = 0.9$, $\beta_2 = 0.999$, $\epsilon = 1e−8$. We set warm-up steps to 500 for all methods in Table 1 and 7. We set the warm-up steps to 0 for all methods in Table 2 and 9, which based on our experiments, improved the results for all methods.

---

[9]https://data.quora.com/First-Quora-Dataset-Release-Question-Pairs

Table 6: Performance of all methods on the tasks in GLUE for different values of hyper-parameters. For each method, we report the total number of parameters across all tasks and the number of parameters that are trained for each task as a multiple and proportion of T5$_{\text{BASE}}$ model [3]. For MNLI, we report accuracy on the matched validation set. For MRPC and QQP, we report accuracy and F1. For STS-B, we report Pearson and Spearman correlation coefficients. For CoLA, we report Matthews correlation. For all other tasks, we report accuracy. Bold fonts indicate the best results in each block.

| Method | #Total params | Trained params / per task | CoLA | SST-2 | MRPC | QQP | STS-B | MNLI | QNLI | RTE | Avg |
|---|---|---|---|---|---|---|---|---|---|---|---|
| INTRINSIC-SAID ($d'$=0.4K) | 1.001 | 0.0002% | 0.0 | 92.55 | 78.43/85.62 | 90.25/87.19 | 90.43/90.66 | 69.93 | 89.31 | 58.99 | 75.76 |
| INTRINSIC-SAID ($d'$=1.4K) | 1.001 | 0.0006% | 52.40 | 93.35 | **89.22/92.41** | 90.44/87.31 | 89.86/90.23 | 82.01 | 93.12 | 67.63 | 84.36 |
| INTRINSIC-SAID ($d'$=2.5K) | 1.001 | 0.0011% | 45.78 | 93.92 | 89.22/92.20 | **90.43/87.37** | 90.32/90.90 | 82.86 | 93.17 | 64.03 | 83.65 |
| INTRINSIC-SAID ($d'$=10K) | 1.001 | 0.0045% | 56.13 | 93.58 | 88.73/91.99 | 90.34/87.18 | **90.63/90.99** | 84.84 | 93.36 | **71.22** | 85.36 |
| INTRINSIC-SAID ($d'$=20K) | 1.001 | 0.0090% | **58.69** | **94.15** | 88.24/91.78 | 90.28/87.13 | 90.06/90.45 | **85.23** | **93.39** | 70.50 | **85.45** |
| PHM-ADAPTER ($n$=4) | 1.018 | 0.239% | 59.21 | 93.69 | 87.25/90.91 | 90.23/86.99 | 90.55/90.73 | 85.93 | **93.04** | 69.78 | 85.30 |
| PHM-ADAPTER ($n$=8) | 1.011 | 0.160% | **61.84** | 93.58 | 91.18/93.57 | **90.25/87.08** | **90.74/91.07** | 85.74 | 92.93 | 70.50 | 86.23 |
| PHM-ADAPTER ($n$=12) | 1.013 | 0.179% | 57.35 | **94.50** | **91.67/93.86** | 90.25/87.05 | 90.45/90.84 | **85.97** | 92.92 | **75.54** | **86.40** |
| COMPACTER ($n$=4) | 1.004 | 0.073% | **63.75** | 93.00 | 89.22/92.31 | **90.23/87.03** | 90.31/90.74 | 85.61 | 92.88 | **77.70** | **86.62** |
| COMPACTER ($n$=8) | 1.004 | 0.073% | 61.78 | **93.81** | 90.20/93.10 | **90.23/87.03** | 90.16/90.44 | **85.78** | 93.08 | 74.10 | 86.34 |
| COMPACTER ($n$=12) | 1.004 | 0.073% | 61.38 | 93.69 | **91.18/93.71** | 90.11/86.88 | **90.53/90.98** | 85.76 | **93.12** | 70.50 | 86.17 |
| COMPACTER++ ($n$=4) | 1.002 | 0.047% | 61.27 | 93.81 | 90.69/93.33 | 90.17/86.93 | **90.46/90.93** | 85.71 | **93.08** | **74.82** | **86.47** |
| COMPACTER++ ($n$=8) | 1.002 | 0.047% | 62.79 | 92.55 | 88.24/91.95 | 90.16/86.94 | 90.43/90.78 | 85.36 | 92.82 | 73.38 | 85.95 |
| COMPACTER++ ($n$=12) | 1.002 | 0.048% | **63.01** | **93.92** | **91.18/93.75** | **90.23/87.01** | 90.40/90.65 | 85.46 | 92.88 | 71.22 | 86.34 |

# B    Impact of Hyper-parameters

In this section, we study the impact of hyper-parameters for each method reported in Table 1. We report the results in Table 6.

**Impact of dimension ($d'$) on INTRINSIC-SAID** Increasing the dimension $d'$ for INTRINSIC-SAID method often improves results. Though, as discussed in [14], $d'$ is task-dependent so needs to be tuned for every new dataset to achieve optimal performance.

**Impact of $n$ on PHM-ADAPTER:** Table 6 shows the results for varying values of $n = \{4, 8, 12\}$. We experiment with adapters of bottleneck size $d \in \{24, 48, 96\}$.

For the T5$_{\text{BASE}}$ model with $k = 768$, the condition $kd > n^4$ discussed in §4 is partially satisfied for $d = 24$ and $n = 4, 8$ and fully satisfied for $d \in \{48, 96\}$ and $n = 4, 8, 12$. Note that this condition is satisfied for larger versions of the T5 model, i.e., T5-large (770 million parameters, $k = 1024$), T5-3B (2.8 billion parameters, $k = 1024$), and T5-11B (11 billion parameters, $k = 1024$) with adapter hidden size $d \in \{24, 32, 48, 96\}$ and $n = 2, 4, 8, 12$. Due to the huge computational costs of training these models, we could not run experiments on such a large scale. Nevertheless, we observe substantial parameter reduction using PHM-ADAPTER.

In Table 6, we report the number of parameters for $d = 24$ for all methods. Compared to ADAPTER, PHM-ADAPTER with $n = 8$ reduces the parameters substantially by $5.2\times$.

**Impact of $n$ on COMPACTER:** For COMPACTER and COMPACTER++, we observe that the number of trainable parameters is almost constant across different values of $n$. This is due to the fact that the number of trainable parameters in layernorms (LN) and biases (B) in each LPHM layer make up a high proportion of parameters for our methods. For instance for $n = 4$, for COMPACTER with 0.073% of trainable parameters, LN and B make up 28.49% and 23.51% respectively of its trainable parameters; for COMPACTER++ with 0.047% of trainable parameters, LN and B make up 44.01% and 18.15% respectively of its parameters; while for PHM-ADAPTER with 0.239% of trainable parameters, LN and B make up only 8.63% and 7.12% respectively of its parameters. Consequently, simply removing biases from adapters, and exploring ideas of training language models without layer normalizations [35] can be promising directions on reducing parameters further, which we leave to future work.

COMPACTER has more than an order of magnitude fewer parameters compared to ADAPTER, with a parameter reduction at a remarkable rate of $11.4\times$. COMPACTER++ even reduces the parameters further by $17.7\times$ in total.

## C   Results with Fine-tuning the Output Layer

Table 7 shows the results for the methods in Table 1 with fine-tuning the output layer. The parameters of the output layer dominate the parameters of each method and thus reduce the relative parameter savings. The standard adapter obtains the largest improvement in performance when fine-tuning the output layer compared to the results in Table 1. In contrast, our proposed methods perform well with or without fine-tuning the output layer.

Table 7: Performance of all methods on the tasks in GLUE, where the output layer is tuned. For each method, we report the total number of parameters across all tasks and the percentage of parameters that are trained for each task as a multiple and proportion of T5$_{\text{BASE}}$ model [3]. For MNLI, we report accuracy on the matched validation set. For MRPC and QQP, we report accuracy and F1. For STS-B, we report Pearson and Spearman correlation coefficients. For CoLA, we report Matthews correlation. For all other tasks, we report accuracy.   Bold fonts indicate the best results in each block.

| Method | #Total params | Trained params / per task | CoLA | SST-2 | MRPC | QQP | STS-B | MNLI | QNLI | RTE | Avg |
|---|---|---|---|---|---|---|---|---|---|---|---|
| *Baselines* | | | | | | | | | | | |
| ADAPTER | 1.065 | 11.89% | 61.80 | **94.15** | 88.24/91.67 | **90.27/87.05** | 91.51/91.71 | 86.02 | 92.64 | **76.26** | **86.48** |
| PFEIFFER-ADAPTER | 1.032 | 11.49% | **64.76** | 93.58 | 87.75/91.58 | 90.16/87.17 | 91.21/91.50 | 86.16 | **93.30** | 73.38 | 86.41 |
| ADAPTERDROP | 1.038 | 11.56% | 61.67 | 93.69 | 84.80/89.20 | 90.14/87.17 | 90.92/91.34 | **86.24** | 93.23 | 73.38 | 85.62 |
| ADAPTER-LOWRANK | 1.004 | 11.13% | 62.82 | 93.81 | 88.73/91.99 | 90.34/87.19 | 90.51/90.58 | 85.81 | 92.93 | 74.82 | 86.32 |
| BITFIT | 1.010 | 11.19% | 57.13 | **94.15** | **89.71/92.78** | 90.07/87.02 | 90.91/91.22 | 85.34 | 93.06 | 68.35 | 85.43 |
| *Our Proposed Methods* | | | | | | | | | | | |
| PHM-ADAPTER ($n{=}4$) | 1.017 | 11.30% | 62.79 | 93.58 | **89.22/92.41** | 90.23/87.01 | 90.61/90.81 | **86.06** | 92.95 | **75.54** | **86.47** |
| PHM-ADAPTER ($n{=}8$) | 1.011 | 11.22% | 61.24 | **94.38** | 88.73/91.99 | 90.28/87.08 | 90.53/90.98 | 85.94 | **93.03** | 73.38 | 86.14 |
| PHM-ADAPTER ($n{=}12$) | 1.013 | 11.24% | **65.25** | 93.69 | 88.73/92.04 | **90.34/87.16** | **90.75/90.89** | 85.74 | 92.92 | 72.66 | 86.38 |
| COMPACTER ($n{=}4$) | 1.004 | 11.13% | **61.27** | 93.58 | 88.24/91.67 | 90.25/87.08 | **90.67/91.02** | 85.82 | 92.92 | 73.38 | 85.99 |
| COMPACTER ($n{=}8$) | 1.004 | 11.13% | 60.31 | **93.81** | 89.71/92.63 | 90.23/87.02 | 90.49/90.85 | 85.19 | **93.08** | 71.94 | 85.93 |
| COMPACTER ($n{=}12$) | 1.004 | 11.13% | 59.25 | 93.12 | **91.18/93.75** | **90.31/87.16** | 90.37/90.82 | 85.33 | 92.97 | **75.54** | 86.35 |
| COMPACTER++ ($n{=}4$) | 1.002 | 11.11% | **64.28** | **94.27** | 90.20/92.96 | **90.23/87.04** | 90.27/90.61 | 85.80 | **92.97** | 73.38 | **86.55** |
| COMPACTER++ ($n{=}8$) | 1.002 | 11.11% | 63.78 | 93.58 | **90.20/93.01** | 90.19/87.02 | 90.12/90.56 | 85.57 | 92.84 | 70.50 | 86.12 |
| COMPACTER++ ($n{=}12$) | 1.002 | 11.11% | 62.05 | 93.23 | 87.75/91.58 | 90.19/86.97 | 90.08/90.48 | 85.52 | 92.75 | **79.86** | 86.41 |

## D   Results on SUPERGLUE

Table 8 shows the performance of our proposed methods on SUPERGLUE for different values of $n$. We include the learning rate obtaining the best validation performance for all methods reported in Table 2 in Table 11.

Table 8: Performance of our proposed methods on the tasks in SUPERGLUE for different values of $n$. For each method, we report the total number of parameters across all tasks and the percentage of parameters that are trained for each task as a multiple and proportion of T5$_{\text{BASE}}$ model [3]. For CB, we report accuracy and F1. For MultiRC, we report F1 over all answer-options (F1$_a$) and exact match of each question's set of answers (EM) [19]. For ReCoRD, we report F1 and EM scores. For all other tasks, we report accuracy.   Bold fonts indicate the best results in each block.

| Method | #Total params | Trained params / per task | BoolQ | CB | COPA | MultiRC | ReCoRD | WiC | Avg |
|---|---|---|---|---|---|---|---|---|---|
| PHM-ADAPTER ($n{=}4$) | 1.013 | 0.24% | **80.31** | **85.71/73.52** | 44.0 | **71.99/51.65** | **74.62/73.60** | 67.40 | **69.20** |
| PHM-ADAPTER ($n{=}8$) | 1.008 | 0.160% | 79.39 | 82.14/69.87 | 44.0 | 71.49/50.77 | 74.46/73.48 | 67.71 | 68.15 |
| PHM-ADAPTER ($n{=}12$) | 1.009 | 0.179% | 79.33 | 78.57/75.43 | **52.0** | 70.48/50.66 | 74.14/73.14 | **68.65** | 69.16 |
| COMPACTER ($n{=}4$) | 1.003 | 0.073% | **79.88** | 89.29/82.51 | 42.0 | **71.87/51.98** | **74.64/73.59** | 65.83 | 70.18 |
| COMPACTER ($n{=}8$) | 1.003 | 0.073% | 79.57 | 85.71/80.06 | **56.0** | 70.75/49.67 | 74.56/73.57 | **70.85** | 71.19 |
| COMPACTER ($n{=}12$) | 1.003 | 0.073% | 78.59 | **96.43/87.44** | 48.0 | 70.80/49.67 | 74.49/73.54 | 65.20 | **71.57** |
| COMPACTER++ ($n{=}4$) | 1.002 | 0.047% | **79.94** | 85.71/80.06 | 50.0 | 72.16/50.33 | 74.63/73.60 | **68.34** | 70.53 |
| COMPACTER++ ($n{=}8$) | 1.002 | 0.047% | 78.23 | 82.14/70.87 | 48.0 | **71.61/51.43** | **74.62/73.64** | 67.71 | 68.69 |
| COMPACTER++ ($n{=}12$) | 1.002 | 0.048% | 78.84 | **92.86/84.96** | **52.0** | 70.68/50.99 | 74.55/73.50 | 68.03 | **71.82** |

# E   Impact of Model Size

Table 9 shows the results of methods using T5$_{SMALL}$ (60M parameters) on GLUE. For all adapter-based methods, we experiment with adapters of bottleneck size of $\{16, 32, 64\}$. For our methods, we experiment with $n = \{4, 8, 16\}$.

All parameter-efficient fine-tuning methods are performing worse than full fine-tuning with this small model size. This is in contrast to the results of Table 1 and 2, where some parameter-efficient fine-tuning methods were able to perform on par or outperform full fine-tuning with the larger model size of T5$_{BASE}$ (222M parameters). Among all methods, adapters, and our proposed methods perform the best. We report the learning rate performing the best on the validation set of each method in Table 10.

Table 9: Performance of all methods on the tasks in GLUE. For each method, we report the total number of parameters across all tasks and the percentage of parameters that are trained for each task as a multiple and proportion of T5$_{SMALL}$ model [3]. For MNLI, we report accuracy on the matched validation set. For MRPC and QQP, we report accuracy and F1. For STS-B, we report Pearson and Spearman correlation coefficients. For CoLA, we report Matthews correlation. For all other tasks, we report accuracy. Bold fonts indicate the best results in each block. We repeat the experiments marked with ∗ multiple times for different seeds, but they were not successful.

| Method | #Total params | Trained params / per task | CoLA | SST-2 | MRPC | QQP | STS-B | MNLI | QNLI | RTE | Avg |
|---|---|---|---|---|---|---|---|---|---|---|---|
| *Baselines* | | | | | | | | | | | |
| T5$_{SMALL}$ | 8×1 | 100% | **46.90** | **91.74** | 87.25/90.97 | **90.07/86.68** | 88.75/89.20 | **82.20** | **90.59** | **65.47** | **82.71** |
| ADAPTER | 1.054 | 0.698% | 36.88 | 90.83 | **88.73/91.93** | 88.09/84.06 | 88.98/89.34 | 80.50 | 89.75 | 62.59 | 81.06 |
| ADAPTERDROP | 1.009 | 0.139% | 34.73 | 89.91 | 83.33/88.36 | 87.96/83.89 | 88.73/88.80 | 79.33 | 89.86 | 61.87 | 79.71 |
| PFEIFFER-ADAPTER | 1.027 | 0.363% | 38.86 | 90.48 | 85.78/89.90 | 87.82/84.26 | **89.24/89.56** | 80.63 | 89.84 | 57.55 | 80.36 |
| ADAPTER-LOWRANK | 1.005 | 0.090% | 40.55 | 90.60 | 84.80/89.20 | 88.01/83.98 | 88.04/88.27 | 79.92 | 89.95 | 61.15 | 80.41 |
| PROMPT TUNING-R | 1.007 | 0.085% | 0.0* | 86.35 | 68.14/81.05 | 87.48/83.91 | 87.35/87.87 | 76.27 | 88.49 | 50.36 | 72.48 |
| PROMPT TUNING-T | 1.007 | 0.085% | 0.0* | 79.59 | 71.08/82.18 | 87.76/83.55 | 87.48/87.76 | 74.65 | 89.02 | 57.55 | 72.78 |
| BITFIT | 1.015 | 0.190% | 25.59 | 90.48 | 84.80/89.42 | 88.01/83.77 | 87.58/87.89 | 78.15 | 88.94 | 63.31 | 78.90 |
| INTRINSIC-SAID | 1.003 | 0.033% | 0.0* | 90.25 | 84.80/89.05 | 88.07/84.00 | 87.81/88.08 | 79.02 | 89.90 | 52.52 | 75.77 |
| *Our Proposed Methods* | | | | | | | | | | | |
| PHM-ADAPTER ($n=4$) | 1.015 | 0.216% | 40.08 | 90.60 | **86.27/90.21** | 88.26/84.25 | 89.56/89.88 | 80.73 | **90.10** | 60.43 | 80.94 |
| PHM-ADAPTER ($n=8$) | 1.011 | 0.170% | 37.85 | 90.48 | 82.84/87.72 | 88.08/84.07 | 89.07/89.46 | 80.68 | 89.64 | 61.87 | 80.16 |
| PHM-ADAPTER ($n=16$) | 1.031 | 0.414% | 36.27 | **90.83** | 83.82/88.34 | 88.03/84.02 | 87.94/88.44 | 80.04 | 89.95 | 58.99 | 79.70 |
| COMPACTER ($n=4$) | 1.005 | 0.090% | **44.65** | 89.45 | 84.80/89.20 | 88.00/83.96 | 88.19/88.47 | 79.54 | 89.66 | 64.03 | 80.90 |
| COMPACTER ($n=8$) | 1.005 | 0.091% | 42.90 | 89.56 | 84.31/89.12 | 88.01/83.95 | 88.51/88.79 | 79.60 | 89.68 | **66.19** | **80.97** |
| COMPACTER ($n=16$) | 1.006 | 0.097% | 40.12 | 89.22 | 85.29/89.86 | 88.08/84.06 | 89.28/89.60 | 79.87 | 89.71 | 59.71 | 80.44 |
| COMPACTER++ ($n=4$) | 1.003 | 0.059% | 39.89 | 90.37 | 84.31/89.26 | 88.04/83.99 | 88.69/88.98 | 79.45 | 89.05 | 63.31 | 80.49 |
| COMPACTER++ ($n=8$) | 1.003 | 0.059% | 34.98 | 90.37 | 83.82/88.50 | 88.02/83.99 | 88.87/89.30 | 79.39 | 89.57 | 64.03 | 80.08 |
| COMPACTER++ ($n=16$) | 1.003 | 0.065% | 37.54 | 89.79 | 85.78/90.90 | 88.01/83.96 | 88.93/89.30 | 79.35 | 89.40 | 64.75 | 80.61 |

Table 10: Selected learning rates for all methods with T5$_{SMALL}$

| Method | Learning rate |
|---|---|
| T5$_{SMALL}$ | 3e−4 |
| ADAPTER | 3e−3 |
| PFEIFFER-ADAPTER | 3e−4 |
| ADAPTERDROP | 3e−3 |
| ADAPTER-LOWRANK | 3e−3 |
| PROMPT TUNING-R | 3e−2 |
| PROMPT TUNING-T | 3e−1 |
| INTRINSIC-SAID | 3e−2 |
| BITFIT | 3e−3 |
| PHM-ADAPTER | 3e−3 |
| COMPACTER | 3e−3 |
| COMPACTER++ | 3e−3 |

Table 11: Selected learning rates for all methods with T5$_{BASE}$ on SUPERGLUE.

| Method | Learning rate |
|---|---|
| T5$_{BASE}$ | 3e−4 |
| ADAPTER | 3e−4 |
| PFEIFFER-ADAPTER | 3e−4 |
| ADAPTERDROP | 3e−4 |
| ADAPTER-LOWRANK | 3e−3 |
| PROMPT TUNING-R | 3e−2 |
| PROMPT TUNING-T | 3e−1 |
| BITFIT | 3e−4 |
| INTRINSIC-SAID | 3e−3 |
| PHM-ADAPTER | 3e−3 |
| COMPACTER | 3e−3 |
| COMPACTER++ | 3e−3 |