# OpenReview forum: "Compacter: Efficient Low-Rank Hypercomplex Adapter Layers"
_NeurIPS.cc/2021/Conference — NeurIPS 2021 Poster_

### Official Review · Reviewer_Ndst · 2021-07-16

**Rating:** 6
**Confidence:** 5

**Summary:**

The authors presented an efficient parameterization of task-specific adapters for parameter-efficient multi-task transfer learning with large pre-trained language models based on transformer architecture.


**Main Review:**

In general, the method is well described and demonstrated with pre-trained T5-base on GLUE.  The idea is novel and its motivations from prior work are clearly stated.  That said, I do have a few criticisms that I hope the authors would address.

- **Model capacity**.  It is not clear whether Compacter is applicable to transformer LMs of different sizes, since only T5-base is demonstrated.  When hidden dimensions are smaller, are low-rank adapters still adequate?
- **Task difficulty**.  It is not clear how the applicability of Compacter vary with difficulties of tasks.  Out of the GLUE tasks studied, how does the effectiveness of Compacter correlate with, say, word-order sensitivity (arXiv:2012.15180)?  What about even harder tasks and longer sequences?  Any results on SQuAD?
- **Parameter efficiency**.  Though the authors conducted a comprehensive comparison on parameter efficiency, some other approaches might be even more efficient in parameters--for example, it has been shown that BERT can be fine-tuned for any GLUE task by setting a small fraction of pre-trained weights to zero, i.e. _sparsification as fine-tuning_ (arXiv:2004.14129), in which case only the row/column indices of the small fraction of weight entries are necessary to store in addition to the pre-trained weights.
- **Practical significance**.  The approach of adding task-specific adapters to pre-trained LMs, however parameter and/or compute efficient the adapters are, still leads to a computational cost no less than that of the pre-trained network.  Pre-trained SOTA transformer models are grossly overparameterized and, just like other large NNs, can be compressed (by sparsification, quantization and/or distillation) for efficient inference in practice.  Thus, the adapter approach might not be as desirable as simultaneous compression/fine-tuning approaches such as _sparsification as fine-tuning_.


**Time Spent Reviewing:**

2

---

> ### Author Response · Authors · 2021-08-06
> **Response to Reviewer #4**
>
> > “Model capacity. It is not clear whether Compacter is applicable to transformer LMs of different sizes, since only T5-base is demonstrated. When hidden dimensions are smaller, are low-rank adapters still adequate?”
>
> Adapters are a general efficient finetuning method. As shown in their original work of Houlsby et al. (2019), and the T5 paper, one can apply them to different language model sizes.
> We provided a comparison for T5-base, a 220 million parameter model. This is the largest model, which could train on given our computational limits. We did not use t5-small as our base model as we thought it may be too small and reviewers might prefer to see the results for t5-base. We are happy to provide some experiments/analysis with t5-small to showcase the effectiveness of the method on t5-small as well.
>
> > “Task difficulty. It is not clear how the applicability of Compacter vary with difficulties of tasks. Out of the GLUE tasks studied, how does the effectiveness of Compacter correlate with, say, word-order sensitivity (arXiv:2012.15180)? What about even harder tasks and longer sequences? Any results on SQuAD?”
>
> As confirmed with R1, R2 we provided solid experiments on GLUE, which is a well-known benchmark in NLP covering diversity of the tasks. We believe our paper is complete as is, but we are happy to provide experiments on more benchmarks possibly some tasks from SuperGLUE or questions answering datasets like SQUAD.
>
> > “Effectiveness of compactor to word-order sensitivity”
>
> Our focus on this work is to provide and study a parameter-efficient finetuning method. Studying the effectiveness of parameter-efficient finetuning methods on word-order sensitivity is an interesting direction, but it does not fit in the scope of this work.
>
> > “Parameter efficiency. Though the authors conducted a comprehensive comparison on parameter efficiency, some other approaches might be even more efficient in parameters--for example, it has been shown that BERT can be fine-tuned for any GLUE task by setting a small fraction of pre-trained weights to zero, i.e. sparsification as fine-tuning (arXiv:2004.14129), in which case only the row/column indices of the small fraction of weight entries are necessary to store in addition to the pre-trained weights.”
>
> Thanks for pointing us to this interesting work. However, this work does not address training time/memory efficiency and addresses an orthogonal issue of inference time efficiency, meaning that one needs to train a mask during the training and pay the whole memory and training time costs of large-scale language finetuning models. After obtaining the mask, they require a second iteration to make the network sparse and during inference time only they can make the network more efficient. However, our goal in this work, is to propose a parameter-efficient finetuning method with the goal to save substantially on memory and training time costs.
>
>
>
> > “Practical significance: The approach of adding task-specific adapters to pre-trained LMs, however parameter and/or compute efficient the adapters are, still leads to a computational cost no less than that of the pre-trained network.”
>
> We provided efficiency evaluation in section 5.4, comparing with the first row (which is full finetuning), our proposed solution, compactor++, provides -41.94% saving on memory and -26.51% saving on training time. This is because adapters will only backpropagate a small number of parameters in large-scale language models by freezing the majority of parameters, and this can substantially help in terms of memory usage and faster training time.
>
> > “Pre-trained SOTA transformer models are grossly overparameterized and, just like other large NNs, can be compressed (by sparsification, quantization and/or distillation) for efficient inference in practice. Thus, the adapter approach might not be as desirable as simultaneous compression/fine-tuning approaches such as sparsification as fine-tuning.”
>
> This submission addresses training time/memory efficiency. However,  the reviewer’s suggestions are related to testing time efficiency, which we are not studying in this work.    These are orthogonal issues. For sparsification/quantization/distillation methods, one needs to pay the costs for full finetuning and  then in the second stage apply such specifications/quantization. While the final results can be efficient at inference time, such methods cannot help with the training time and memory usage during the training phase, while our method provides advantages in this regard.

---

### Official Review · Reviewer_zZfs · 2021-07-16

**Rating:** 5
**Confidence:** 4

**Summary:**

This paper propose an parameter-efficient fine-tuning method called Compactor by inserting a low-rank matrix decomposition adapter. Experiments support the claim.

**Limitations And Societal Impact:**

Yes.

**Main Review:**

# Pros
Compactor basically combines the low-rank decomposition with adapter. The writing is clear and idea is very easy to follow. Experimental results seem reasonable.

# Cons
1. Limited novelty: the idea is not new to the community. As in the reference [24], the low-rank decomposition has already demonstrated effectiveness in batch ensemble learning. Extending it to multitask fine-tuning is natural and intuitive.

2. Weak results: authors only show results on GLUE benchmarks, which is not convincing. What about the performance on SuperGLUE, low-resource tasks and text generation?



**Time Spent Reviewing:**

2

---

> ### Author Response · Authors · 2021-08-06
> **Response to Reviewer #3**
>
> > “Limited novelty: the idea is not new to the community. As in the reference [24], the low-rank decomposition has already demonstrated effectiveness in batch ensemble learning. Extending it to multitask fine-tuning is natural and intuitive.”
>
> We believe there might be a misunderstanding here. In [24], the authors study the impact of low-rank decomposition on ensemble learning. While in our work, we are not studying ensemble learning and we have a very different method and setting, building on intuitions of multiple very recent pieces of work on intrinsic dimensionality of language models [14] (ACL, 2021), and hypercomplex layers [17] (ICLR, 2021). We proposed a solution for efficient finetuning of large scale language models. Our hypercomplex low-rank methods for modeling adapter layers have not been proposed, nor studied in any of the prior work. Also, we do not consider multi-task finetuning in this paper, and provided experiments are on single-task tuning with compact adapter layers (compacter).
>
> > “Weak results: authors only show results on GLUE benchmarks, which is not convincing.
>
> We evaluate the method thoroughly on the GLUE benchmark, which is a well known and standard benchmark in NLP covering a diverse range of the such as sentiment analysis (SST-2), linguistic acceptability (CoLA), paraphrase detection (MRPC, QQP), sentence similarity (STS-B), multiple natural language inference datasets, in which some of them have been converted from other domains such as coreference resolution or question answering (MNLI, QNLI, RTE). As confirmed by R1, R2, we have provided extensive analysis and thorough experimental results as well as a comparison with recent approaches, showing that our method outperforms recent parameter-efficient finetuning methods and full finetuning.
>
>
> > “What about the performance on SuperGLUE, low-resource tasks and text generation?”
> > “low-resource tasks”:
>
> we provided experimental results on low-resource tasks in section 5.5 by subsampling GLUE benchmark tasks for varying sample sizes (100, 500, 1000, 2000, 4000).
>
> > “SuperGLUE”:
>
> As confirmed by R1, R2 the paper comes with thorough analysis and experimental results as is, with comparison with recent parameter-efficient fineutning methods. We are happy to provide further experiments on SuperGLUE or possibly question answering benchmarks as additional evidence in the camera ready version.
>
> > “text generation”:
>
> The focus of our work is on NLU tasks, while extension to NLG tasks is an interesting future direction, this is out of scope of this work and we leave it to future work.

---

> > ### Comment · Reviewer_zZfs · 2021-08-29
> > **Appreciate your efforts**
> >
> > Dear Authors,
> >
> > Thanks for your rebuttal. After reading the rebuttal and other reviews, I am inclined to retain the current score. Compared with [24], novelty and originality are still my major concerns. The improvements are not considered as significant.

---

### Official Review · Reviewer_CErC · 2021-07-17

**Rating:** 6
**Confidence:** 4

**Summary:**

This paper develops a new parameter-efficient adapter layer, which extends the parameterized hypercomplex multiplication layers by sharing part of parameters across layers and decomposing adapter-specific parameters with two low-rank weights. Experimental results on the GLUE benchmark prove that the proposed approach outperforms other parameter-efficient finetuning methods and obtains performance on par with full finetuning.

**Limitations And Societal Impact:**

Please see the detailed comments.

**Main Review:**

Strengths:
1. The authors propose a novel parameter-efficient adapter layer to adapt the large-scale language model.
2. The proposed method requires dramatically fewer parameters than adapters and finetuning, while outperforming other parameter-efficient fine-tuning methods.

Weaknesses:
1. The novelty of this paper is limited, which is a straightforward combination of parameterized hypercomplex multiplication layers, sharing information across adapters, and low-rank parameterization. These methods are widely known.

Detailed Comments:

In general, this paper is easy to follow, the idea is very straightforward, and the parameter efficiency is analyzed in detail. Experimental results on the GLUE benchmark verify the effectiveness of the proposed method in terms of the parameter-efficient and accuracy. But the proposed method is not novel enough to me. It is a straightforward combination of some previous techniques, including parameterized hypercomplex multiplication layers, sharing information across adapters, and low-rank parameterization. The low-rank parameterization method actually could further reduce parameters of the original adapter layer or the AdapterFusion method. Adding these experimental comparisons could make the experiment more convincing. In addition, the authors only verify the effectiveness of the proposed method on NLU tasks. Actually, the proposed method can directly apply to harder NLG tasks. I am curious whether this method achieves high performance on downstream NLG tasks with such few parameters.


**Time Spent Reviewing:**

6

---

> ### Author Response · Authors · 2021-08-06
> **Response to Reviewer #2**
>
> > Strengths “The authors propose a novel parameter-efficient adapter layer to adapt the large-scale language model.”
>
> > “Weaknesses “The novelty of this paper is limited, which is a straightforward combination of parameterized hypercomplex multiplication layers, sharing information across adapters, and low-rank parameterization. These methods are widely known”
>
> We disagree that the combination of these methods is straightforward and note that such a combination has not been studied in any prior work. Aghajanyan et al. (ACL 2021) provide the intuition that large-scale language models can have a lower intrinsic dimension. Their proposed method though, as shown in the paper, is very inefficient in terms of memory and training time. On the other hand, Aston et al. (ICLR 2021) for the first time proposed hypercomplex multiplication layers.  While we base our work on these recent work, as described in the paper, section 4, our proposed solution is not a straightforward combination of their methods, as it successfully leverages many intuitions for efficiently fine-tuning large language models and additionally our method decreases the complexity from O(kd/n) (hypercomplex layers applied to adapters) further to O(k/n+d/n). This is a substantial reduction in terms of number of parameters compared to any recent prior work. We are also the first to propose an efficient formulation for low-rank approximation of adapters and we show through extensive analysis that our proposed method outperforms recently proposed methods.
>
> > “The low-rank parameterization method actually could further reduce parameters of the original adapter layer or the AdapterFusion method. Adding these experimental comparisons could make the experiment more convincing.“
>
> Thank you for the suggestion. We have experimented with low-rank adapters and found that it is not expressive enough to achieve strong performance. In particular, low-rank adapters with “0.07253%” total trainable parameters obtain only average performance of 85.78 on average on GLUE, while compactor++ obtains 86.47 on average with only 0.047% of parameters (roughly half of the parameters). Please see the detailed results of low-rank adapters below:
>
>
> | Method | Trainable parameters| CoLA     | SST-2       | MRPC     |QQP        | STS-B  | MNLI | QNLI | RTE | Average |
> |:---         |    :----:                         |   :----:       |    :----:       |     :----:     | :----:        | :----:      | :----:   | :----:   | :----:   |   ---:|
> |Low-Rank Adapters | 0.073%    |  57.17  | 94.15  | 90.69/93.38  | 90.32/87.22 |  90.78/91.1   |  85.62 | 93.39  |  69.78 | 85.78 |
> | Compacter (n=4)    |  0.073%   | 63.75   | 93.00  |  89.22/92.31 |  90.23/87.03 |  90.31/90.74 |  85.61 |  92.88 | 77.70 |86.62  |
> | Compacter++ (n=4) | 0.047%   | 61.27   |  93.81 |  90.69/93.33 |  90.17/86.93 |90.46/90.93  |  85.71 | 93.08 |  74.82 | 86.47 |
>
> We will include these results in the revised version of the paper.
>
> > “In addition, the authors only verify the effectiveness of the proposed method on NLU tasks. Actually, the proposed method can directly apply to harder NLG tasks. I am curious whether this method achieves high performance on downstream NLG tasks with such few parameters.”
>
> As you mentioned, we have provided a detailed analysis and “experimental results on the GLUE benchmark which verifies the effectiveness of the proposed method in terms of the parameter-efficient and accuracy”.  We therefore believe the paper has a complete set of experiments, although we will include more experiments on possibly SUPERGLUE and question answering tasks. Since our focus is on NLU tasks, further study on NLG tasks can be an interesting future direction, we leave this to future work as this is out of scope of this study.

---

### Official Review · Reviewer_s8Se · 2021-07-21

**Rating:** 9
**Confidence:** 4

**Summary:**

This paper propose COMPACTER, a parameter-efficient fine-tuning method  for large-scale language models. On GLUE datasets, COMPACTER only requires training 0.047% parameters without loss of accuracy, and moreover it outperforms standard fine-tuning in low-resource settings.


**Limitations And Societal Impact:**

Yes

**Main Review:**

* Pros
  * Method is novel and interesting. COMPACTER builds on ideas from adapters, low-rank methods as well as recent hypercomplex multiplication layers.
  * Experiments and analysis are solid. It compares against a large number of baselines, and did thorough analysis on memory and training time.
  *  Results are strong — it can achieve only 0.05% without loss of accuracy on GLUE, outperforming other parameter-efficient fine-tuning methods.
   * The paper is well-written.
  * Parameter-efficient NLP method is an important direction.
* Cons
  * It will be great to also validate on datasets other than GLUE, e.g. Question Answering.
* Missing references
  * Guo et al, Parameter-efficient transfer learning with Diff Pruning

**Time Spent Reviewing:**

1

---

> ### Author Response · Authors · 2021-08-06
> **Response to Reviewer #1**
>
> > “Missing references”
>
> Thanks a lot for highlighting this paper. We will include it in the camera-ready version.
>
> > Cons “It will be great to also validate on datasets other than GLUE, e.g. Question Answering.”
>
> As you mentioned, the paper comes with a thorough analysis/experiments as is and we have provided a solid comparison with recent approaches. We will include an evaluation on question answering datasets (like  SQUAD) or possibly some of the tasks in the SuperGLUE benchmark to provide further evidence to the camera ready version.

---

### Author Response · Authors · 2021-08-06
**General Response to all reviewers**

We would like to thank all reviewers for their insightful comments.
We are glad that R1, R2, and R4 find the method novel and interesting, experiments and analysis are solid (R1), authors conducted a comprehensive comparison on parameter efficiency (R4), results are strong (R1), provided experimental results prove that the proposed approach outperforms other parameter-efficient finetuning methods and full finetuning (R2), we compare against a large number of baselines (R1), and did thorough analysis on memory and training time (R1), experiments support the claim (R3). The paper is easy to follow (R2) and the method is well described (R4), the motivations from prior work are clearly stated (R4).  The paper’s topic is an important direction (R1).
We briefly address the two main criticisms mentioned in the reviews (see detailed replies in the individual responses).

> “Limited novelty. The method combines existing ideas.”

R1, R2, and R4 consider our work novel. We propose a method which leverages recent methods, but is not a straightforward combination of them. Our method is designed to leverage recent insights related to the intrinsic dimensionality, information captured during transfer, as well as observed redundancies in transferred and fine-tuned layers of large language models. These insights motivate our use of general techniques such as hypercomplex multiplication and decomposing layers into shared and layer-specific components.  For example, one component of the hypercomplex layer is shared between all layers and another one is approximated with a low-rank matrix.  Theoretically, our formulation improves parameter efficiency substantially to O(k/n+d/n) compared to both adapters’ O(kd) and hypercomplex layers applied to adapters’ O(kd/n). Empirically, it outperforms other comparison methods on GLUE and is Pareto-optimal on a performance/parameter efficiency curve (Figure 1). As such, we believe our proposed method forms a novel and non-trivial contribution for the setting of fine-tuning large language models as well as a meaningful improvement compared to prior work.

> “Only evaluated on GLUE.”

We provide a thorough analysis and solid experiments on GLUE (R1, R2, R3, R4), which is a standard benchmark in NLP that consists of a diverse set of different tasks. Most methods we compare against have, in fact, only evaluated on this benchmark. We additionally perform experiments under low-resource conditions and provide an efficiency comparison of recent methods. The provided results are strong (R1), and demonstrated experimental results prove that the proposed approach outperforms other parameter-efficient finetuning methods and full finetuning (R2). However, we will also evaluate our method on QA tasks and/or SuperGLUE tasks in the camera-ready version of the paper to provide further evidence of the effectiveness of our method.

---

> ### Author Response · Authors · 2021-08-10
> **Results on SuperGLUE**
>
> Please find our obtained results on the SuperGLUE benchmark. As the original test sets are not publicly available, we follow Zhang et al. [26]: For larger datasets, we split off 1k samples from the training set as the validation set, while we use the original validation data as the test set. For datasets with fewer than 10k samples, we divide the original validation set in half, using one half for validation and the other for testing.
>
> Following [1], we tried with maximum length of 256 for SuperGLUE (For record, following [1] and [2] we tried with maximum length of 512). Compacter and Compacter++ obtain strong gains on the SuperGLUE datasets, improving the performance on average by 1.52 and 1.88 respectively over the full fine-tuning results, with only finetuning 0.073% and 0.048% of parameters respectively.
>
> | Method  | Trainable parameters | BoolQ   | CB  |CoPA    | WiC  | ReCoRD   | MultiRC | Average|
> | :---------- | :------------- |:------------- | :----------: | -----------: |---------:|----------:| ----------:| ----------:|
> | Full finetuning| 100% |      **81.1** |  85.71/78.21 |   52.0 |   70.22 | 74.28/73.37 |68.71/45.93|  69.95 |
> | Compacter|     **0.073%**    | 79.57 | 85.71/80.06 |            **56.0** |          **70.85** |      **74.56/73.57**|  **71.51/51.43**| **71.47**|
> | Compacter++|    **0.047%**   |      78.84 | **92.86/84.96** |            52.0 |          68.03 |      **74.57/73.55** |   70.68/50.99|**71.83** |
>
> Caption: Following [3], for CB, we report accuracy and F1, for ReCoRD we report max (over all mentions) token-level F1 and exact match (EM), for MultiRC we report F1 and EM, for all the other datasets, we report the accuracy.The best result in each column is bolded.

---

> > ### Author Response · Authors · 2021-08-17
> > **Finalizing the results on SuperGLUE**
> >
> > We now included the results on Superglue for all methods. Compacter and Compacter++ obtain substantial improvement over finetuning, and improve the performance on average by 1.52 and 1.88 respectively with only finetuning 0.073% and 0.048% of parameters respectively.

---

### Decision · Program_Chairs · 2021-09-27

**Decision:**

Accept (Poster)

**Comment:**

This paper address efficient finetuning of pertrained models and investigate applications in NLP domains. The idea is to insert a parameter compact layer called Compacter in the pretrained models. By further parametering the Compacter layers with low-rank representation, its achieves a better trade-off between model performance, trained parameters and memory footprint compared with existing methods.

The idea is quite simple and the techniques are well explored in different scenarios by existing works. Thus most reviewers consider the technical contribution to be incremental. However, it demonstrates very good results in NLP pretrained models. Some reviewers were not satisfied with the limited experiments provided in the original submission. In the rebuttal, the authors was able to provide more results in a more complex data and model. And the results are in consistent with the results provided in the paper. The reviewers are generally satisfied with the rebuttal. Considering the technical limitation and the support from the reviewers, I recommend a weak accept and the authors to incorporate the new results and other comments from the reviewers into the revision.